# Health and Psychological Concerns of Communities Affected by Per- and Poly-Fluoroalkyl Substances: The Case of Residents Living in the Orange Area of the Veneto Region

**DOI:** 10.3390/ijerph20227056

**Published:** 2023-11-12

**Authors:** Marialuisa Menegatto, Adriano Zamperini

**Affiliations:** FISPPA Department, University of Padova, Via Venezia 14, 35131 Padova, Italy

**Keywords:** health ostracism, chronical contamination exposure, human biomonitoring, chemical trespass, personal violation, social comparison, social psychology

## Abstract

Residents of an extensive area of the Veneto Region (Italy) face one of the largest technological disasters due to per- and polyfluoroalkyl substances (PFAS). On the basis of a risk gradient of contamination, the affected territories were divided into 4 areas: Red (of maximum exposure, where a human biomonitoring programme (HBM) was activated), Orange, Yellow, and Green. This article presents a case study of residents who live in the Orange Area, the second area in terms of contamination, excluded from the HBM. Semi-structured interviews were conducted with 17 residents engaged in promoting a legal procedure to exercise their right to know. Grounded theory and a thematic analysis method were used. Overall, the findings revealed that experiencing contamination causes a negative psychosocial impact on the residents’ lives; difficulty accessing information; living with uncertainty, caused by the lack of institutional and health support and medical consultation; a sense of abandonment; difficulty managing preventive and protective actions; and the deterioration of relationships, on the basis of the social comparison with residents of the Red Area, to whom HBM was granted and where the concept of health ostracism has emerged. This study demonstrated that biomonitoring may help reduce discomfort in the case of contamination by informing people of their chemical exposure.

## 1. Introduction

For decades, residents of an extensive territory of the Veneto Region have been exposed to high levels of per- and polyfluoroalkyl substances (PFAS) by way of contaminated ground (bore), town water, and local produce, and other exposure pathways. The Veneto Region is located in the north-eastern part of Italy and it is famous for its historical capital city: Venice. PFASs comprise a diverse family of synthetic organic chemicals that have been used in a variety of industrial and consumer products worldwide since the mid-twentieth century. Since their genesis, PFAS have been detected in ambient environments, wildlife, and human serum around the globe [1,2,3]. Due to their astonishing physical–chemical properties, such as being friction-reducing, oil- and water-repellent, and temperature-resistant, PFAS are highly resistant to biological, chemical, and photochemical degradation and are of serious concern to human and environmental health [4]. Mounting scientific and clinical evidence has found that adverse physical and psychological health outcomes across all life stages are associated with PFAS exposure, including reduced kidney function, metabolic syndrome, thyroid disruption, cancer, adverse pregnancy outcomes, and chronic distress [5,6,7].

PFAS contamination in the Veneto Region was identified in a series of European and Italian research projects conducted from 2006 onwards [7]. These results revealed massive contamination in the territories of three provinces, namely, Vicenza, Padova, and Verona. The pollution plume extended not only to surface waters but also to deep-fault waters, from which water is extracted for human consumption, to irrigate crops, and for livestock consumption, implying potential risks for humans. It was confirmed that the health of hundreds of thousands of people was at risk. In addition, it was possible to attribute this environmental disaster to the Miteni Group, a factory located in the Trissino Municipality in the province of Vicenza, which has produced PFAS since the 1960s. In December 2016, after the drinking water supply chain had been reconstructed by the water service companies that manage supply, the Veneto Region divided the affected territories into four areas based on a risk gradient [8,9] (see Figure 1). The Red Area, the area of maximum health exposure and impact, comprises the municipalities where public drinking water, well water, surface water, and groundwater are all contaminated by PFAS and residents are highly exposed. The Orange Area comprises the municipalities where PFAS contamination was detected only in private well water, surface water, and groundwater; no contamination of the public drinking water was identified; and residents were moderately exposed. The Yellow Area is a zone of special attention in which a control system for the environmental networks for surface and groundwater, including irrigation and drinking water, was put in place. Lastly, in the Green Area, PFAS is only found in environmental matrices, requiring further monitoring and studies.

Therefore, a free regional health surveillance programme for residents of only the Red Area was launched in 2016 in response to public and private concerns, with the aim of facilitating the prevention, early diagnosis, and treatment of some of the residents’ chronic disorders. The programme includes human biomonitoring (HBM) to investigate the PFAS levels in their bodies and thus provide essential scientific data to design appropriate preventive actions, public health surveillance programmes [11,12], and, where high PFAS levels were found, clinical observations and other laboratory tests. For the Yellow and Green Areas, where municipalities were only slightly impacted by PFAS, a monitoring programme was established to better understand the extent of the exposure.

## 2. The Orange Area and the Right to Know

Currently, the Orange Area includes 12 municipalities. The contamination originates (epicentre) from the Trissino Municipality, where the Miteni Company is situated (Figure 1). At present, the company is on trial for its responsibility in causing this technological disaster. Even though the Orange Area ranks second in terms of PFAS exposure, the authorities have not only excluded its residents from the free regional health surveillance programme but also rejected individual requests for voluntary HBM, even if the residents are willing to bear the expenses. The reasons were based on this scientific rationale: the drinking water contamination did not affect the Orange Area through the aqueduct network, so the population was exposed to a lower level of PFAS than the Red Area [13]. For residents, this is like being denied twice; they cannot access the free collective health programme, even if they ask for it. In the latter case, it does not enable their family doctors to provide them with necessary medical prescriptions for HBM.

These Orange Area communities must deal with diverse challenges, which, in turn, can interact to exacerbate the impacts on the physical and psychological health [7] of their residents and act as cumulative factors [14]. Residents are aware of how PFAS contamination negatively impacts human health; they know the PFAS levels in the blood of residents in the Red Area; they know the food contamination levels in the Red Area; they are aware of experiencing chronic environmental contamination (CEC) [15,16]; and some of them live near the epicentre of the contamination of the Miteni Company, whose site has not yet been cleaned. However, they do not know, in fact, they are forbidden from knowing the levels of PFAS in their bodies, and they have been denied access to healthcare or, at least, that access has been limited. Other events have contributed to increasing concerns among residents of the Orange Area. For example, in 2013, the Vicenza Municipality closed the Scaligeri well, located in the west area, and promptly isolated it from the water network because the level of perfluoroalkyl substances was above the legal limit [17]. Some neighbouring municipalities also drew water from this well, but their exposure to contamination remains unknown, despite being potentially dangerous. Additionally, in 2021, the public swimming pool of the Creazzo Municipality in the Orange Area was closed because high levels of PFAS were found. In the same year, after a four-year tug-of-war between residents and the Veneto Region, disclosure was made of part of a study carried out by the National Health Institute, sponsored by the Veneto Region, on the contribution of food contamination to total human exposure in the Red Area, assessed via a food monitoring campaign between 2016 and 2017. The results showed widespread food contamination that was higher for the animal matrix than for the vegetable matrix [18,19]. Among Orange-Area residents, these events have provoked many more questions than among Red-Area residents, as they have not yet been answered, such as ‘what’s in our bodies?’, ‘what’s in our land?’, ‘can our disease be caused by PFAS?’, and ‘are our children contaminated?’. Local environmental hazards can plunge entire communities into social, political, and economic chaos [20,21] and, in this specific case, leave residents’ existence suspended in time amid uncertainty and indeterminacy. Further, the interaction of environmental and social stressors has been described as ‘double jeopardy’ [22].

In this scenario, chronic threats are posed, and conflicts have arisen between the community and the Veneto Region around the right to know [23]. The people in the Orange Area, exposed to toxic contaminants, have been passed over by health surveillance programmes. Consequently, the Veneto Region’s decision was defied by a group of Orange-Area residents through the procedure allowed by Italian law. They undertook collective action to exercise their right to know by accessing HBM because they were unable to implement prevention strategies and were deprived of the opportunity to know the damage they had suffered. The recognition of the right to know allows citizens to make decisions on how to protect their health and the health of loved ones and to exercise their autonomy and right to self-determination [23]. According to Hadden [24], information allows citizens to make decisions, changing the balance of power. In other words, information gives people the power to make decisions, implement individual changes, and request systemic changes in the management of the phenomenon of environmental contamination.

## 3. The Role of Human Biomonitoring in Chemical Trespass

Previous psychosocial studies on communities exposed to PFAS contamination in the Red Area found that residents had a feeling of personal violation [7,25] when they discovered the amount of PFAS in their blood after taking HBM tests. This violation extended beyond their physical health to have a negative psychological impact due to deep concerns about their health and safety.

HBM is widely used in environmental health science to detect and understand the health implications of chemical trespass in people’s bodies. It is an important instrument for assessing the human concentration and internal body burden of chemicals, usually through analyses of biomarkers in bodily fluids [26], facilitating informed decisions on health protection within the risk governance cycle [27].

Since the mid-20th century, we have seen an increasing and rapid expansion of biomonitoring population studies in various countries and continents, such as the United States, Europe, and Canada [28], to measure a growing number of chemicals in ever larger groups of participants. A wide range of environmental exposure has been investigated, from historic pollution, such as lead [29,30], to synthetic chemicals, including pesticides [12,31,32,33], to new emerging chemicals [12,34,35]. Hence, HBM is rapidly becoming a key means of providing a scientific basis for prevention via exposure reduction and motivating action. For example, in the past, HBM was used to investigate lead levels in children living in contaminated homes to measure their exposure and monitor prevention strategies [36]. In Europe and California, biomonitoring data from breast milk were used to encourage the phase-out of certain polybrominated diphenyl ethers (PBDEs), which are flame retardants used in electronic equipment, furniture, and other products [37].

The public is actively involved to varying degrees. For example, HBM is also used in community-based studies to respond to local concerns about environmental health risks or by civil society organisations [38,39]. Many residents of the Orange Area consider HBM a way to help them identify PFAS levels that may be harmful to their health, take protective steps to reduce the potential health risk for adults and children, and address potential hazards.

HBM studies commonly report results in two ways: by interpreting aggregated statistical data or by generating personal results regarding individual concentration, often accompanied by some comparison with reference or population values. In the latter case, ethical issues emerge associated with the communication of personal results to study participants. There is no universal consensus or common practice in this regard [40,41]; therefore, the debate turns to an ethical dilemma concerning whether or not to communicate personal results to study participants [38]. Traditionally, most HBM studies have adopted a clinical ethical perspective by communicating the results only at the aggregate level if concentrations are exceeded or if there are serious risks and health has clearly been compromised [23,42]. In the framework of community-based participatory research (CBPR), however, results of preventative or precautionary significance should be disseminated to participants to motivate behaviour change, raise awareness of environmental health risks, and increase trust between participants and scientists [42,43]. Furthermore, participants generally want to know their results, and the principle of autonomy includes the right to know (or, conversely, the right not to know) as a basis for self-determination in acting on research results [23]. Studies on ‘the exposure experience’ [44,45] have shown that participants are not passive receivers but active actors who, in addition to receiving results, ask questions and make interpretations according to their past experiences, attitudes, and perceptions [46,47]. Ultimately, HBM has important implications for environmental justice [48] and disadvantaged communities.

In this article, we investigate the chronic environmental exposure experience (CEC) of residents in the Orange Area, the right not to know, their exclusion from HBM, and their fight to know.

## 4. Materials and Methods

The present study is the first part of a mixed-methods community health resilience (CHR) research project, currently in progress, based at the University of Padova (Italy) (FISPPA Department) that aims to investigate the psychosocial consequences of PFAS exposure for affected populations in the Veneto Region of Italy. After collecting data from residents living in the Red Area [7,25], we designed a secondary study on the experience of residents who lived in the Orange Area, in particular analysing their sense of control over their condition through a small group of citizens who legally exercised their right to know their personal blood PFAS levels. According to the study design adopted [7], in this phase, we selected in-depth interviews as the most suitable method to capture the concerns, opinions, perceptions, emotions, and experiences of locals and provide an opportunity to explore topics that will inform the development of further studies using quantitative methods [49]. The investigation was conducted in two phases. The first phase took place in 2022, immediately after the first contact. The topics covered in these interviews included the following broad areas: the discovery of PFAS contamination; the experience of being in the Orange Area rather than the Red Area; perceptions and understanding of PFAS exposure; the feeling of not knowing; the reasons behind the decision to join legal procedures to exercise the right to know; the psychological costs of the fight; and the meaning of HBM. The second phase took place in 2023. On 11 January 2023, when we were in the data analysis phase, the Veneto Region decided to allow residents of Orange Area municipalities to access HBM [50], responding to the expressed need. If, on the one hand, the Veneto Region recognised residents’ right to know through HBM, on the other hand, it required them to share the health costs by imposing a tax of 90 Euros per person. Therefore, we included this event in our study to assess the opinions of residents on the decision taken by the Veneto Region.

### 4.1. Recruitment and Data Collection

In our study, we involved a group of residents of the Orange Area engaged in promoting the cause of the right to know the levels of PFAS in their bodies. Our first contact with them was made at the beginning of May 2022, when two meetings were organised to create a connection between researchers and participants and to implement a communication strategy to let potential participants know about the study.

Seventeen group members (*n* = 9 female, *n* = 8 male) joined the CHR study phase with a focus on the Orange Area. The general inclusion criteria for the study included being a resident in the Orange Area, which was considered the second most contaminated area and a source of PFAS contamination; having submitted for a review, through a lawyer, the decision of the Veneto Region to exclude Orange-Area residents from HBM; not having previously undergone a PFAS medical test; and being of legal age. Respondents came from different municipalities in the Orange Area. As the COVID-19 pandemic was ongoing in Italy, and according to the precautionary measures in place, we used a videoconferencing platform to collect interview data. Advances in communication technologies offer new opportunities to conduct qualitative research [51,52,53], and online methods can replicate traditional methods, including in-person interviews [54]. This technology helped us reach a larger area when recruiting participants, building on a strategy we previously tried [7]. Two researchers (M.M. and A.Z.) interviewed the participants in two phases.

The first phase took place between July and September 2022. All in-depth interviews lasted between 40 and 60 minutes and were audio- and video-recorded and transcribed verbatim; participants reaffirmed their consent verbally and in writing before the online interviews were conducted and completed a brief socio-demographic questionnaire. The second phase took place in February 2023, when we asked participants to integrate this study considering the provision decided by the Veneto Region. The participants received a series of questions via email, and everyone answered via email.

This study followed the American Psychological Association Ethical Principles of Psychologists and Code of Conduct and the principles of the Declaration of Helsinki. It was approved by the University of Padova Ethics Committee (Protocol Number 1D4BA484CC28FCDA6984C4F21E59DEA6).

### 4.2. Data Analysis

We used thematic analysis [55,56], a qualitative method that provides rich, detailed, and complex accounts of data. The qualitative software package Atlas.ti was used to assist in data management. After familiarising ourselves with the content of the transcripts, we constructed a mixed inductive and deductive coding matrix through the selection of text parts, named quotations; thereafter, codes were grouped into categories, and categories into themes. A thematic structure was developed with themes and sub-themes. Two independent coders and a third external coder analysed the data.

## 5. Results

### 5.1. Participant Demographics

Seventeen residents (*n* = 9 female, *n* = 8 male) of the Orange Area participated in the study. They ranged in age from 48 to 70 (Mean = 55.6; SD = 5.67); all participants were married. Eight had received a middle-level education (47%). All were parents and had between two and a maximum of three children. The children of the participants ranged in age from 6 to 35 (Mean = 19.85; SD = 7.35). All participants came from municipalities located in the Orange Area, where they had resided for an average of 34.5 years, ranging from 8 to 70 years. All owned the houses in which they lived, most of which had gardens and vegetable gardens (see Table 1).

### 5.2. Themes and Sub-Themes

All findings were organised to reflect three major inductive and deductive interrelated themes emerging from our analysis: the discovery of the contamination, health ostracism, and the fight to know. Seven sub-themes were associated with the emerging themes (see Table 2).

#### 5.2.1. Discovery of the Contamination

This emerging theme addressed when and how residents involved in the study discovered PFAS contamination. Eight residents learned about it in 2013–2014, when the news began to circulate in the communities through local media articles, municipality monthly information sheets that warned citizens to stop using water from private wells, emails sent by groups to which residents were subscribed, such as *Gruppi di Acquisto Solidale* (GAS; ethical purchasing collectives), groups for synergistic garden courses, or word of mouth among friends. In this first phase, people were not too worried about the implications of the contamination. In fact, the residents’ territories were subject to frequent pollution events, and these facts mitigated perceptions and the resulting concerns. As one resident said, “First, I thought about local pollution, as had happened other times, so my attention has waned”. A resident also reported that in the 1970s, an important chemical pollution incident led to the intervention of water tankers from the nearby U.S. army base, after which there was an institutional vacuum. 

However, the alarming news came mainly from the Lonigo Municipality, which was a considerable distance from their own area, while reassurance messages came from the local municipalities. As these interviewees argued: 

“The news came from Lonigo Municipality, an area from which I felt rather distant”.

“Water is under control. It is a normal situation” or “We have not heard reports of risk or danger to public health”.

The issue strongly re-emerged in 2016 when the Veneto Region divided the impacted territories into different administrative areas, and the health surveillance programme, with calls for HBM in the Red Area, consequently started. At this point, the interviewed residents discovered that they were living in a PFAS-contaminated area designated ‘Orange’, which was being treated differently from that provided for the residents of the Red Area. At the same time, they were hearing incontrovertible news about HBM data from relatives and others in the Red Area, such as the MammeNoPfas (MothersNoPfas), and their public denunciation of their children’s PFAS blood levels was capturing the interest of the national mainstream media. The participants reported an initial reaction of feelings of loss, dismay, and disbelief because they had never imagined that the situation would become so pervasive. For example, some residents said:

“I felt a little bit astonished; I thought someone would take action before we got to this situation”.

“I just kept saying that it was impossible, I never imagined that pollution would continue over the years like this… in disbelief that we are still living with this”.

“I couldn’t believe it; it seemed so crazy to me”.

All the residents declared consequent concerns and feelings of danger because the risk was becoming concrete: 

“At first, I was incredulous, maybe a little bit relieved not to be in the Red Area, but then I understood that the risk of PFAS contamination was real and serious even for the Orange Area”.

“I felt in danger, red or orange is the same because water goes everywhere, we don’t stop it!”.

##### Personal Violation

Although the Orange Area had been declared less dangerous than the Red Area, the participants affirmed that discovering PFAS pollution was a dramatic experience, especially when they thought about the food they had consumed over the period of contamination. The residents interviewed largely bought and consumed local food products because they considered them healthy and reliable. From the moment of the discovery, the participants believed and emphasised that they were in danger from the food because their territories had become contaminated lands where they felt threatened, rather than safe and protective places. In fact, the water used to irrigate fields, vegetable gardens, and orchards or to provide water for animals came from private contaminated wells or rivers, in turn, contaminating the products they had eaten—in their minds, safely—for several years. As these participants reported: 

“They tell us that the problem only refers to groundwater, but are the products of our lands safe?”.

“My neighbour brought us vegetables from his garden watered from his well, for years…”. 

One resident described the feeling of discomfort well: “Knowing that everything produced in these lands is contaminated, I feel a deep discomfort”. Meanwhile, insecurity was the main manifestation of the feeling of personal violation. As this participant said:

“We feel insecure, what have we eaten so far? What water is used for vegetables? For the hens? And the eggs? How do I know? I have many doubts!”.

Some participants stressed the link between insecurity and knowing, as demonstrated by the following quote: “Not knowing what quality of food and water you use makes me feel insecure”. Another participant referred to the PFAS levels in the drinking water: “They tell us that in drinking water, the levels of PFAS are acceptable… but we have to consider the accumulation in our bodies, it only takes a little over years to cause damage!”

In general, the residents tried to deal with this insecurity by adopting sustainable coping strategies for problems, such as checking the origin of food or buying bottled water, despite the negative emotions such actions brought. As this participant explained well:

“Before, to going to the farmer to buy local products was a joyful moment. Now, I must go far away to areas where river or aqueduct water is not used. This is very sad, and penalises everyone, me and the farmer, who suffers enormous damage. It is like someone destroyed your house. The sensation is really that the foundations of our earth have cracked, devastated, with a very bad sensation of that there is no solution to all this”.

##### Loss of Well-Being

This sub-theme emerged when participants noted how the discovery that they were living on contaminated land and the health ostracism negatively impacted their well-being. It was as if they had suffered an assault on their fundamental expectations that care would be taken in an ordered world where normality assumes that citizens are allowed to protect their health. An “absurd barrier” was what one interviewee called the partition of territories by the region, “which prevented them from protecting their own and their families’ health and preventing anything negative from developing”.

This perception of violation passed through the territory, for which there was a particular level of place attachment and which was considered to be somehow marked and defended from the intrusiveness of danger, which has left residents with the problems of loss of spatial control and discomfort, as reflected in the following quotes:

“These territories are our oasis; the idea that they are contaminated it makes me sad”.

“I look at it with despair”.

The emotional experience is nourished by intrusive thoughts: all the interviewees think all the time about the amount of PFAS they may have in their bodies. This rumination involves repetitive and passive thoughts focused on the causes and effects of PFAS contamination. For example, these residents said:

“The thought of the water drives me crazy, although they tell us that everything is ok”.

“There is always the idea of illness… Unfortunately, over the years, we have heard so much about ill people who died of cancer. How do you forget about it?”.

“It is like we are not worthy of well-being”.

Secondly, through the interviews, it became apparent that a major element of discomfort is the loss of the right to tranquillity because they and their loved ones were denied the chance to take preventative healthcare measures:

“Nobody reassures us, no one helps us to take care of ourselves and our family”.

Further, one resident suspected that the pollution was behind his own rare disease, discovered a few months earlier, and another thought the same in connection with his father’s illness. Other participants described some premature deaths among young people or incidents of the spread of breast cancer, saying “They alarm us!”

#### 5.2.2. Health Ostracism

This theme describes the participants’ state of being ignored, excluded, or barred from the public health system due to the persistent denial of access to HBM by the Veneto Region for residents of the Orange Area. When the interviewed residents were asked to describe their experience of living as people barred from the HBM health service, they generally answered that “It is like living in limbo”, a state of suspension in which “I have too many questions and doubts without answers”.

All participants described ways in which their discomfort had been invalidated by the healthcare system. They were not at ease with their primary care physician, whose answers were always vague, or private analysis laboratories due to denied access to blood tests. One participant was specifically told that he could not access blood tests as such tests had been prohibited by the region, while another participant formally requested HBM several times without receiving a response:

“I formally wrote to the region, to the local health service, to my primary care physician, but no one ever answered me”.

This invalidation often led participants to feel hesitant to ask about PFAS contamination. For residents, it seemed a sort of discrimination because this health service was continually denied, even if sought voluntarily, individually, and as a private service. Through interviews, it became apparent that a major element was the invisibility they experienced in the Orange Area, and what they were going through internally, as one resident explained:

“Living in the Orange Area means that you must run, run, and run without reaching the destination because you are missing a piece of paper, that paper where it is clearly written how much PFAS you and your family have in your blood”. 

This institutional non-recognition indicates an experience of exclusion and marginality, as this participant reported: 

“We feel pushed aside compared to residents of the Red Area, who are aware of the risk they are bearing because they have done biomonitoring tests… and each of them is able to make their own decisions”.

The participants also described exclusion from public healthcare services that did not invite them to take any kind of blood test or medical examination to monitor the situation over time, followed by the perception of being second-class citizens.

##### Uncertainty and Health Concerns

Living in a situation of not knowing is difficult. Receiving information about a contaminated environment, as well as about individual diagnostic outcomes, allows people to give a name to their condition and follow a specific protective or therapeutic programme. Even when no effective treatment is available, as in the case of PFAS contamination, obtaining one’s biomonitoring data regarding PFAS levels increases their chances of being able to deal with the contamination. The consequence is a sort of double uncertainty, as expressed by this participant:

“It is a very bad situation, to know that there is something wrong, but we don’t know what it is, we don’t do tests, and we can’t do any checks that inform us. Unfortunately, we can’t even adopt behaviours or to make choices that can somehow protect us”.

The experience of living as if one’s existence was suspended was also mentioned by an interviewee who stated the following: “I fear to remain like this, as in a sanitary and medical uncertainty”. And it becomes even more complex, since the residents interpreted the symptoms and illness from which they suffer, or which affected their loved ones, as being due to PFAS contamination. However, these inferences remain “inexplicable”, as reported, shrouded in doubts and concerns, as they cannot be medically demonstrated to be correlated to the PFAS contamination, as these participants reported:

“I have thyroid problems. They could be congenital and they could also be derived from PFAS exposure. Living in the Orange Area also means not really knowing the danger to which we are exposed”.

“My mother-in-law had pancreatic cancer. My mother had three cancers in a few years”.

The interviewees stated that this lack of objective data generates deep uncertainty: “This means living with uncertainty: can they be linked to the PFAS?” and “We practically live under uncertainty”. For this reason, they explicitly expressed concerns about their health and that of their families and friends. A significant consequence of not receiving objective data is the sense of isolation. Due to the lack of health attention and support, some residents turned to other healthcare professionals, such as physicians who belong to the Italian Association of Doctors for the Environment (ISDE), on their own initiative and privately to make sure that no “sentinel values” were in their bodies.

##### Emotional Experience

The uncertainty characterising the health ostracism due to PFAS contamination in the Orange Area had a negative psychological impact on the residents’ lives. As mentioned above, the interviewees referred to having an explicit ban on accessing HBM as well as difficulties with healthcare services regarding PFAS and with receiving information about any correlated clinical treatment. This, in their opinion, led them to manage the situation autonomously, to be unsupervised, to be unable to take specific protective action, and to be excluded from benefiting from the surveillance programme or other kinds of institutional support. The most common emotions reported were loneliness, due to the feeling of being forgotten by institutions, and of being abandoned, due to the institutional silence on their situation, as this resident stressed:

“We live in an area seriously affected by contamination; nevertheless, it is as if it doesn’t exist, we feel neglected”.

The residents complained of the lack of healthcare support not only because they were denied, but also because the lack of formal information from the institutions has repercussions in their daily lives. PFAS contamination is barely mentioned publicly in the Orange Area; hence, they must often update themselves on what is happening in the Red Area. Furthermore, feeling abandoned by doctors and the healthcare system generated feelings of fear, worry, frustration, discouragement, and anger. This emotional experience is well expressed in these quotes: 

“I am so scared now and for the future. I’m worried for my children, we don’t know how that’s going to turn out, it is as if we are denied a peaceful life”.

“You yell at someone, and no one listens: frustration has become unsustainable”. 

Very often, anger is felt towards institutions, which are accused of abandoning them, inefficiency, a lack of control of industrial plants, and, above all, not allowing residents to take action to protect their health, and the lack of information received, as reported in these following quotes:

“A great anger, the right to defend our health was taken away from us”.

“Anger about not doing something to solve the problem that impacts on people’s life”.

“Anger towards the municipal administration, which has not made any important statement, obviously also towards the local and regional health authority, and the Veneto Region, which should act for the common good”. 

Anger about abandonment involves the erosion of systemic trust, that is, a feeling of distrust towards institutions that originates from a negative dialectic with them and brings with it the perception that they are not fulfilling their duties towards citizens or ensuring a safe and adequate quality of life. In this specific case, the interviewees also revealed experiences of disappointment and deception due to the injustice suffered, leading, in all the participants, to a loss of trust in the regional political institutions and health authorities. This loss of trust is well reflected in the following quotes:

“I have even less trust in institutions. With the PFAS affair, you do realise that they turned their heads away when they had to act in favour of citizens”.

“In these years, I have completely lost the trust in institutions. I have even started to lose trust in justice. Economic interests are very strong, and it seems that they overcome collective interests”.

Institutional inaction, in a situation where rapid action is required to safeguard and protect citizens, is a key problem related to the loss of trust: “They knew, and they did nothing, and are still doing nothing! I have a basket full of distrust”.

#### 5.2.3. Fight to Know

This third theme emerged in relation to the legal demand of residents to obtain access to HBM. Participants described a series of reasons why they made this choice. The first is because their right to health was denied, where health is related to the presence of some pathologies correlated with PFAS. Then, they intend to fill all inherent doubts, even minimal ones; be able to face uncertainty; and finally, have their rights recognised. The following two quotes express these feelings:

“Denying us the possibility of knowing the PFAS levels we have in our blood is inconceivable. I think it is our right. Therefore, I will try to achieve the results in every fair and legal attempt, even if it involves commitment and fatigue”.

“It should be everyone’s right to be able to know the dangers we are facing and also to know our physical condition”.

In line with this, the interviewed residents clearly expressed the view that the provision of information about PFAS levels in their bodies, as well as health promotion advice to protect their and their loved ones’ health, is a crucial aspect in this case of contamination and is a responsibility of the public healthcare sector. They reported having undertaken an active and constant commitment to collecting signatures, sending emails, and making various formal communications that also involved other local associations, but never received answers. They also highlighted the difficulty of being heard by the Veneto Region and other public institutions, such as the Local Health Unit, that were deaf to their requests. It is particularly clear that the fight to know is a result of implicit health ostracism communicated only indirectly; public institutions continually refuse to acknowledge their citizens. This instance of exclusion involves being passively ignored. For residents, being informed citizens means being able to improve their agency around their health, and their ability to influence their health, and take responsibility for the consequences of their choices, as conveyed in the following quotes:

“Blood tests have been a sort of litmus test. If I have many PFAS, it means that something is wrong and that somewhere I absorbed them! If I have low levels, it means that I can continue the same lifestyle”.

“Knowing our health situation, we could decide to move away; if we don’t have it, we can’t decide”.

Here, the participants’ right to know is considered a valuable asset to improve residents’ health status to find the best ways to deal with the contamination problem. The residents interpret the right to now as an indication of the realisation of a larger right, that is, of health and quality of life, and it is not seen in the simple one-directional clinical delivery of data, but in citizens exercising their ability to become active agents.

##### Corrosive Communities

This sub-theme refers to the division and fragmentation generated by the PFAS affair within the Orange Area itself. Specifically, another important dividing line has been created not only between the Orange and Red Areas, but also between residents within the Orange Area and their respective personal attitudes towards PFAS contamination. A clear picture has emerged: on the one hand, some residents want to know and want to ‘go all the way’; on the other hand, there are residents described as being little involved in the issues, indifferent to the contamination, and little- or unaware of the situation in the Orange Area. In general, all participants reported that talking about PFAS with the latter group of citizens is very difficult. As two residents stated:

“Unfortunately, my perception is that most people prefer to be ostriches and ignore the problem. Even I can hardly speak with my friends and acquaintances, they go against all reasonable evidence”.

“I warned my neighbours about some PFAS initiatives, then, getting no answer, I didn’t re-warn them”.

Additionally, the perceived indifference of their community members represents a factor of discomfort, as one interview reported:

“We don’t feel community support, or communion with the rest of our community, which is equally affected by PFAS, is something that pains me… The bitterness of feeling alone”.

The deterioration of relationships with fellow citizens, neighbours, friends, and colleagues was reported by the participants. They specifically stressed that at the community level, it is difficult to discuss PFAS, and stated that, in this context, they often felt stigmatised, as community members treated them as ‘knuckleheads’ or crazy people until they suffered from silent ostracism, even from members of associations with whom they shared environmental concerns. As one participant stated:

“I was very disappointed by the members of the association to which I belong. I always knew that the PFAS issue is complicated and a very long struggle, but they distanced themselves from me. At one point, I felt abandoned”.

##### Active Minorities

Participants described how the PFAS issue contributed to increasing their interest in their contaminated area and in making a greater commitment to local associations, describing themselves as the “few sensitive active residents worried” about their health and that of their unborn children, bearers of a desire for hope and change. As this resident stated: 

“This is not a personal commitment, because I could say: I drink bottled water. No. This problem is related to the future of our country, our people, our children, etc.… In sum, the quality of our life, present and future”.

In addition to legal demands initiated by a lawyer, participants also initiated participation in PFAS-themed initiatives or events. Some residents reported being actively engaged in local institutional groups, such as a local party civic lists, and others in self-organised associations, such as ‘Citizens ZeroPFAS’. Although these groups operate on different levels, as one participant said, “both are very important to act through different levels”.

Furthermore, some residents are part of GAS, that is, groups engaged in a form of critical consumption based on knowledge of the quality and production methods of the products they buy. They declared that they also kept themselves informed through other more active formal associations or informal activist groups, such as MammeNoPfas in the Red Area.

The process of becoming more active and being part of active minorities together with other residents and citizens was reported by the participants as a protective factor that allows them to share the problem and restores their sense of community and support. In fact, two participants said:

“It’s not only my problem, but I’m also sharing a bit of the situation of everyone”.

“In the end, everyone tries to contribute because the more information we find, the more we can protect ourselves, understand the situation, know and support each other. The knowledge of one of us is the knowledge of all”.

Finally, blended support, in person and online through the creation of WhatsApp groups or through Facebook, was referred to as vital to deal with the situation of feeling alone. Mutual support was described as invaluable. For five participants, active commitment was identified as a shared project within their families or a project jointly undertaken by couples. However, emotional barriers to commitment have also been reported. In some cases, the discomfort resulting from over-engagement or overwork causes excessive stress, especially for mothers, who can struggle to balance work, home, and family.

##### Lack of Full and Inclusive Healthcare

When, in January 2023, the Veneto Region agreed to let residents of the Orange Area access HBM, the news was initially greeted by participants with considerable enthusiasm. Various reactions ensued when they suddenly learned about the decision. Positive emotions about achieving an important victory included satisfaction, excitement, happiness, joy, amazement, relief, pride, and hope, as represented in the following quotes:

“Initially, the news surprised me. By now, I thought that there was nothing more to do, and the authorities would keep silent and wait for everything to end, out of exhaustion, in oblivion. I founded hope”.

“The immediate reaction was of victory because it is a long-term, exhausting, and expensive battle conducted with sacrifice”.

However, this positive reaction soon changed to negative reactions as the initial enthusiasm waned. This experience is consistent with the awareness that the right to know had been partially recognised. In fact, the residents found that they must pay 90 Euros per person to access HBM, which led to some unpleasant emotions. First, participants expressed the perception that the Veneto Region must be joking, considering that blood tests are completely free in the Red Area. Second, they argued that if a co-payment of care costs is necessary, the amount stated by the Veneto Region is too high, especially for citizens with low incomes or families with several members, and it will discourage them. Anger, disappointment, and distrust are the main common emotions expressed by participants. The experience of moving from a positive- to a negative-reaction experience is well expressed in the following quotes:

“Initially, I was very happy, but in the following days, compared to what I thought was a victory, it has become a mockery”.

“In a second moment, I felt angry because the Veneto Region had proposed unfair conditions to us”.

“After a few days, excitement gave way to disappointment! I appreciate the openness of the Veneto Region, but I don’t agree with shared costs. Health is a right and not a privilege, especially in the case of a major contamination caused by others. Therefore, human biomonitoring must be accessible to all citizens”.

“What makes me sore is that it is my right to know what level of contamination I have. I’m not responsible for the poisoning. First, the Veneto Region denied me a blood test, and then charged me for it! I’m ever more distrustful of the institutions that should protect our health”.

Furthermore, as a resident noted, the decision of the Veneto Region did not consider the various categories of residents, especially the most vulnerable population group, effectively excluding them: 

“So, therefore, I am perplexed. It seems this is an individual undifferentiated choice… A planned healthcare pathway doesn’t emerge. For instance, general practitioners are not involved! They must show us the healthcare route as well as be a support in the event of need. This is particularly serious for weaker social categories, such as pregnant or breastfeeding women”.

Another factor that emerged from the interviewees and contributed to their negative stance was a lack of information. Participants complained that the Veneto Region did not provide information to all residents; there was no collective call or information about procedures. As one interviewed resident said:

“No information or concrete guidance on how to perform human biomonitoring was provided by the Veneto Region. Additionally, in many parts of the Orange Area, the information may not have arrived”. 

This latter possibility, together with the high cost and risk of unclear procedures, led to non-adherence by much of the population. As one resident reported:

“I wish public health institutions would promote biomonitoring to the population. Including the weaker social categories. On the contrary, this initiative is not publicly known. What we need is a wide-ranging information campaign. It is like when they make a new law and claim that everyone knows about it. Everyone must know! It’s a disgrace!”. 

In conclusion, the interviewed participants perceived a sort of minimal concession by the Veneto Region to stop the legal procedure, but that it had not taken full responsibility for this recognition: “Once again, showing bad faith”, a resident stated, showing “a general disinterest in taking care of communities, a total indifference to the citizens’ demands emerged. We are a long way from a fair and democratic system”.

## 6. Discussion

It is becoming apparent that the effects of living with environmental contamination can detrimentally affect psychological well-being. These effects may even place a great burden on health and well-being, even if the contamination is classified as of minor risk, and public health decisions and actions can have a considerable negative impact on exposed communities. From this study, it has emerged that, although the Orange Area was declared to have a lower risk of PFAS contamination than the Red Area, the discovery of pollution caused participants to live a dramatic day-to-day experience that has had a considerable negative psychosocial impact on their lives. This result is in line with studies on communities affected by PFAS contamination [7,15,16,57,58,59,60] and human-made disasters [61,62,63].

Through a detailed analysis of the interviews, a central aspect that emerges as one of many issues that have repercussions in daily life and in the relational dimension is health ostracism, introduced in the section addressing emerging themes. Based on social comparison theory [64], the interviewed residents referred to the prohibition by the Veneto Region of accessing HBM and the difficulties in obtaining access to PFAS medical consultation or health support.

Based on our study of PFAS-contaminated areas, we believe that this perception is due to social comparison [64], that is, how the interviewed residents compared themselves to the out-group of Red-Area citizens to whom HBM was granted. A key concept emerging is that of a lateral [65] or parallel comparison [66] to others who are seen to be at a similar level in coping with problems. In fact, the common problems were dealing with PFAS contamination and the measures adopted by the Veneto Region to protect exposed citizens, to whatever extent, and to help guarantee the population’s access to healthcare in terms of prevention, diagnosis, and safety. As a result, they perceived that, in comparison with the residents of the Red Area, they have the status of second-class citizens.

The experience of being ignored, excluded, or barred from the public health system and the persistent denial of access to HBM were described as living in a double-sacrifice zone. The first sacrifice is represented by the negative health consequences of contamination, and the second is being excluded from the possibility of knowing how to deal with the exposure. This aspect has repercussions for residents’ well-being, as well as for their daily lives, because it is institutions and the healthcare system that, above all, invalidate their discomfort. Our analysis of the interviews indicates the difficulty of obtaining any evidence or medical confirmation of their status as contaminated or non-contaminated people. The impossibility of finding objective data that can clarify the level of PFAS exposure exposes residents to another form of pain, namely, that of invisibility. The interviews show how invisibility emerges strongly from their experiences because it exposes them to the risk of seeing their discomfort constantly underestimated and delegitimised.

However, several aspects identified by the participants contributed to this negative experience. One of these was personal violation. After the initial alarm due to the discovery of contamination with consequent concerns and feelings of danger, the risk became concrete, especially in the form of the local food they had consumed daily for years. Although the public institution addressed the contamination problem by stating that there was no danger posed by PFAS in drinking water, reassuring residents through public discourse, the participants experienced a serious risk of being personally contaminated through other sources. Additionally, limited or absent information about the PFAS pollution levels in the Orange Area contributed to negatively cementing the link between insecurity and knowing. The absence of knowledge and information about the health consequences of PFAS contamination could have caused worry and confusion in residents [7,58]. For these reasons, while in the Red Area, residents could know their PFAS body levels through HBM, residents of the Orange Area were obliged to remain in ignorance of their contamination.

HBM provides new worldwide techniques in environmental health to detect and understand the health implications of chemical trespass in people’s bodies, allowing people to play an important role in the interpretation of data to make decisions and take action [23]. The ‘Toxic ignorance’ [23] that characterised the public health approach to contamination in the Orange Area has had a negative impact on the participants’ experiences because it caused deep uncertainty about the origin and causes of some symptoms or illnesses among family members, and about how to find effective coping strategies. This caused uncertainty about the present and the future. Living an uncertain and indeterminate existence, and perceiving themselves as marginalised second-class citizens, generated in residents a deep sense of abandonment and isolation by increasing feelings of loneliness, frustration, fear, discouragement, anger, and distrust of institutions. Loneliness was also exacerbated by the community division and fragmentation generated by the PFAS issue [59], which, in fact, worsened community relationships. Perceived community indifference, which participants also called silent ostracism, is another factor related to discomfort. This discriminatory experience may also have led to the development of internal stigma, which occurs when people suffering from a certain contaminated condition interiorise damaging public and collective perceptions and accept them as applicable to themselves [59]. These prolonged and chronic stressful experiences can result in increased cumulative health risks, leading the population to a resignation stage characterised by an inability to overcome threatened psychological needs and feelings of alienation, unworthiness, helplessness, and depression.

However, this experience allowed residents to transform their experience into a collective fight, to give a name to their suffering and mission, and to spread information about their conditions. In fact, these participants wanted, at any cost, to know their own PFAS body levels, and this prohibition motivated and inspired them to take action to improve their knowledge of environmental health and demand the ability to exercise their rights.

For all these reasons, it is crucial that residents receive information about the contamination and obtain data on the PFAS levels in their bodies through HBM. This would allow them not only to see their suffering legitimised and be included in specific protective or care pathways, but also to benefit from a form of protection by exercising personal protective choices. In addition, although in January 2023, the Veneto Region allowed them to access HBM, after an initial sense of greater satisfaction, this decision left the participants in a profound state of discouragement when they heard the costs they would have to pay to exercise their right to know. They compared themselves once again to residents of the Red Area, for whom blood tests are completely free. The lack of detailed institutional information persisted in this phase, contributing to their negative stance. Finally, from a psychosocial point of view, the participants continuously compared themselves with residents of the Red Area in the same circumstances as part of the work they carried out to manage their identity problem and define themselves, as Orange-Area citizens, as having different rights. This suggests that the division of the contaminated territories into different coloured areas by the Veneto Region also caused diverse identities among residents.

By better understanding these aspects, future research could help to orient the design of health programmes to incorporate social comparison, especially for guiding care relationships, health communication, and the dissemination of information on health and risk level. Such information could be used to close the gap in citizens’, the public’s, and healthcare professionals’ understanding of the lived experience of contamination, and to help people live a positive life with their negative conditions. As this case study has shown, how orange residents compare their experiences of PFAS contamination to red residents provides more than an indication of how people making social comparisons perceive their condition and how they adjust to it. It also provides a way of helping to inform and support others who might be going through a similar experience of contamination.

## 7. Limitations

The limitation of this study was the low number of the participants interviewed, that is, 17 respondents, since only some residents expressed an interest in taking part. Therefore, we suggest implementing future research aiming at exploring the ‘Orange status of citizens’ experienced by residents suffering from contamination that the regional government considers to be second-level exposure.

## 8. Conclusions

The present study contributes to the literature by providing an account of the experiences of some of the residents exposed to PFAS contamination in the Veneto Region of Italy. The aim was to describe the experience of residents living in the territories of the Orange Area, the second area in terms of PFAS exposure, who experienced exclusion from HBM and a significant negative impact on their well-being.

In the PFAS contamination phenomenon, recognising the residents who are exposed, and thus, at risk of developing pathologies, is very important in terms of public health policy. Our study suggests that this recognition starts from and passes through access to HBM as a tool to know the contamination levels in one’s own body and decide on further actions. In this case, HBM may help reduce discomfort caused by health ostracism by informing people of their chemical exposure, restoring some trust in public institutions, collectively leveraging results to support advocacy that promotes broader health efforts to fully understand the experience of the contaminated population, and promoting more protective actions and regulations. In addition, we believe that offering institutional answers and information under the auspices of interactive communication with residents is also crucial considering that PFAS contamination in the Veneto Region is considered a technological disaster in terms of the size of the contaminated territories, number of citizens exposed, and time. PFAS are considered forever chemicals; they build up in bodies over time and persist in the environment, leading to chronic contamination. During crises and disasters, interactive communication is an important way to build trust and reassure the people involved. Consistent with the wider literature on this topic, our results confirmed that the experience of environmental contamination is stressful, but intervention approaches and public health responses can make a difference in mitigating the impact on a population.

Finally, this study raises questions that might prove useful avenues for future research, for example, examining the PFAS experience of those belonging to marginalised groups, identifying the unique characteristics of their experiences, and investigating the role of HBM and its implications for PFAS-exposed people, to gain better information and effectively tailor solutions to populations’ needs. We hope that knowledge about this emerging worldwide problem will be developed and increased to make PFAS contamination a condition that can be efficiently managed and promote the development of an integrated approach to residents, thereby mitigating discomfort. Public health actions can have an important impact on distress in contaminated communities.

## Figures and Tables

**Figure 1 ijerph-20-07056-f001:**
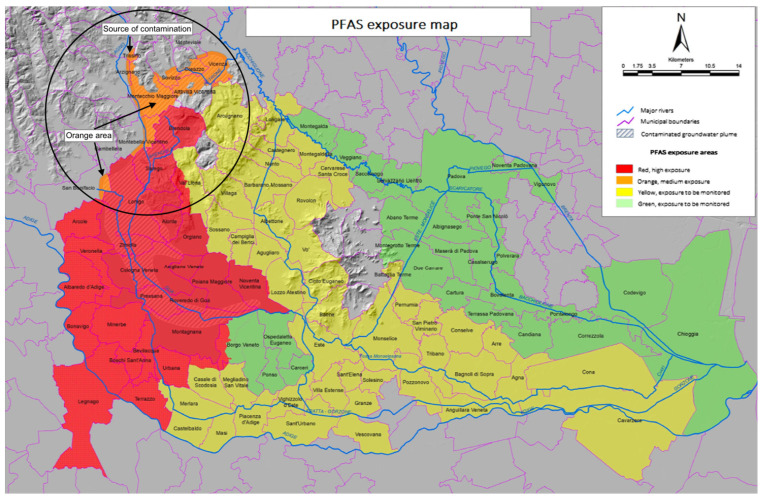
PFAS exposure map indicating the different coloured areas and the municipalities they involve (from reference [10]), including the Orange Area with the source of contamination, annotated by the authors.

**Table 1 ijerph-20-07056-t001:** Characteristics of the study participants (*n* = 17).

Gender	Female	9
	Male	8
Age	Media	55.59
	Range	48–70
	DS	6.67
Family status	Married	17
Children	Under 18	31
	Media	12.52
	Range	1–18
	DS	4.37
	Over 18	34
	Media	26.35
	Range	19–43
	DS	5.74
Education level *	Higher education	7
	Tertiary education	8
	Secondary education	2
Occupation	Employee	11
	Entrepreneur	1
	Accountant	2
	Retired	3
Years of residence	Media	34.47
	Range	8–70
Homeowner	Apartment or home with garden and land	17

* Higher: university/doctoral level; tertiary: middle education; secondary: lower education.

**Table 2 ijerph-20-07056-t002:** The themes and sub-themes that emerged.

Theme	Sub-Theme
Discovery of the contamination	Personal violation
	Loss of well-being
Health ostracism	Uncertainty and health concerns
	Emotional experience
Fight to know	Corrosive communities
	Active minorities
	Lack of full and inclusive healthcare

## Data Availability

The data presented in this study are available upon request from the corresponding author.

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
