# Peer review of "Health and Psychological Concerns of Communities Affected by Per- and Poly-Fluoroalkyl Substances: The Case of Residents Living in the Orange Area of the Veneto Region"

_ijerph, 2023, doi:10.3390/ijerph20227056_

Round 1

Reviewer 1 Report

Comments and Suggestions for Authors

This is a very interesting and important article regarding perceptions, attitudes, and concerns about health and mental impacts from PFAS contamination in a region of Italy.  The paper's methods are sound, and the results and conclusions are appropriate.  The major issue with the paper is that the quality of the English language is poor and therefore some of the points that the manuscript is making could be misunderstood.  I had only the following minor comments on the article:

1) Region of Veneto:  for those readers not familiar with this region, could the authors describe the geography better (e.g., contains the capital Venice, Padua, Verona, etc.)

2) Most likely a translation issue, but were residents of the orange region really "denied access" to biomonitoring, or where they told that they could be biomonitored, but would have to pay their own costs?  I think that distinction could be made more clear.

3) The small sample size is of concern.  Could the authors comment on what they may know about the individuals which refused to participate (i.e. any demographics?).  Can they assess what is the possibility they have a biased sample?

4) Did the authors consider conducting any standardized test/survey of stress on the participants (such as the Perceived Stress Scale?).  What were the barriers to administering such a survey/test?

Comments on the Quality of English Language

Extensive editing for adequate English language should be required before publication; I would recommend a native English speaker review.

Author Response

Thank you very much for you feedback. We appreciate it! In attached file we report the location of our response point by point 

Reviewer 2 Report

Comments and Suggestions for Authors

The manuscript focuses on a case study involving residents in the orange zone, just outside the red zone which was contaminated the most. Residents were interviewed, aimed at advocating for their right to information and legal action. The results highlight the detrimental psychosocial effects of contamination in residents' lives, difficulties in accessing information, uncertainty stemming from it, and the lack of institutional support and medical guidance when compared to residents in the red zone. 

The article is well-structured and serves as a valuable continuation of the authors' previous publication, which involved interviewing residents living in the red zone. 

However, there is a substantial need for extensive English language editing, particularly in the results sections. Several sentences are challenging to understand, making the review process difficult. 

Abstract:

Clarity of Hypothesis and Objectives: The abstract does not clearly state any hypothesis or objectives for the study. 

Methods and Sample Size: The methods employed for gathering the data are briefly described: semi-structured interviews. However, the abstract could benefit from additional details regarding the interview process, like how long were they interviewed and in how many sessions. Additionally, the sample size appears relatively small, with 17 residents, which may limit the generalizability of the findings.

Results: The results indicate that the residents experienced negative psychosocial impact, difficulty to access information, etc. This information is presented clearly.

Discussion and Interpretation: The abstract does not discuss the implications of the findings. It highlights the findings and compares with red area citizens to whom HBM was granted.

Conclusion: It might benefit from a more explicit statement about the overall implications of the study's findings for the field of HBM program.  Overall, the abstract effectively presents the research methods and results, but could benefit from adding implications from their findings and a conclusion.

Introduction:

Background and context: The introduction provides a comprehensive background on the how the region under research was exposed to PFAS substances, source of contamination, reasons for citizens under orange area to be concerned about their exposure and inaccessibility to biomonitoring program, challenges faced by the citizens, importance of human biomonitoring in cases of chemical exposure assessment. All the above context helps readers understand the significance of the study.

Citations and references: The introduction effectively incorporates relevant references to support the claims and context.

Research Gap Identification: the introduction identifies a research gap by investigating the chemical exposure experienced by the citizens in the orange area, and their right to know about their exposure, highlighting their exclusion from HBM program.

Research Objectives: The introduction effectively conveys the primary research objectives, which are to investigate the environmental exposure experience of orange area citizens.

Hypothesis: The introduction does not clearly mention any hypothesis.

Language and clarity: The introduction is generally well-written and clear, but it could benefit from minor grammatical and syntactical adjustments for improved readability.

In summary, the introduction effectively sets the stage for the study by providing relevant background information, identifying research gaps, outlining the methodology, stating clear research objectives, and presenting hypotheses. Incorporating more recent references and making minor language improvements would enhance the overall quality of the introduction.

Materials and methods:

Study Subjects and Ethical Considerations: The section provides clear information about the study subjects, their sources, and ethical considerations, including informed consent and approval by Institutional Review Boards. This demonstrates compliance with ethical guidelines and human subject protections.

Sample collection and size: The section details the collection of data from in-depth interviews conducted over two phases: recorded and transcribed in first phase and over emails in the second phase. The inclusion of both types of interview structures provided a comprehensive assessment. Sample size is relatively small (17 residents), which limits the generalizability of the findings.

Data analysis: The section details how the data was analyzed with constructed thematic coding after content familiarization.

Use of controls: The case study in this article did not use any controls, which could be considered essential to establish a baseline for the exposure assessment.

Overall, the "Materials and methods" section provides a clear and comprehensive description of the experimental procedures and analysis techniques used in the study. Clarifications on sample size and replicates would further improve the section.

Results:

Participant demographics: the study involved 17 residents with 9 females and 8 males. Table 1 is informative is providing details about the participants.

Themes and sub themes: this section provides the themes that emerged from the interviews and categorized the data under three listed themes and seven sub themes. This finding is critical and relevant to the study’s objectives.

Detailed thematic data: These sections include detailed information from the residents’ interviews and their experience about how they feel about being excluded from HBM programs even though they have faced considerable exposure. These sections are overall well organized but would benefit from considerable language editing. Also, it would be beneficial to group the direct verbatims in paragraphs and provide implication of each themes.

Discussion:

Comparative analysis with previous work: Comparison with other work which shows considerable negative psychosocial impact on people who have been exposed to pollution is critical to highlight for the study.

The interview data highlights multiple aspects and concepts about the sacrifices the citizens of orange zone are going through and provided examples of negative experiences which has been causing the citizens to worry or be confused. This section has compared with previous concepts like ‘toxic ignorance’ to detail the discriminatory experiences.

Limitations: the authors acknowledge the low number of samples included in the study. 

Future directions and alternative approaches: The discussion does not touch upon potential future directions. Expanding on these possibilities and their significance would be beneficial.

Clarity and organization: The discussion is well organized and logically structured, making it easy to follow the flow of ideas. However, some paragraphs are quite dense with information, and breaking them into smaller sections could improve readability.

Overall, the discussion effectively summarizes the study's findings, places them in context with related research, and provides insights into the potential implications of the results. Addressing the points mentioned above and providing more context regarding the clinical relevance of these findings would further enhance the discussion.

Conclusion: The conclusion is clear and concise, summarizing the findings without unnecessary elaboration. Overall, the conclusion effectively captures the main takeaways from the study and appropriately addresses the potential significance of the findings. It also highlights the need for further research like examining PFAS experience among marginalized groups of people and evaluating role of HBM programs in developing intervention programs. This conclusion aligns well with the study's objectives and results.  

Comments on the Quality of English Language

There is a substantial need for extensive English language editing, particularly in the results sections. Several sentences are challenging to understand, making the review process difficult. 

Author Response

(The authors gave the same response as above.)

Round 2

Reviewer 2 Report

Comments and Suggestions for Authors

Thank you for making the suggested corrections.